# Optimization of the Emissions Profile of a Marine Propulsion System Using a Shaft Generator with Optimum Tracking-Based Control Scheme

**Joel R. Perez [1,*,†] and Carlos A. Reusser [2,†]** 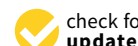

[1] Naval and Maritime Sciences Institute, Faculty of Engineering Sciences, Universidad Austral de Chile, 5111187 Valdivia, Chile

[2] Department of Electronics, Universidad Tecnica Federico Santa Maria, 2390123 Valparaiso, Chile; carlos.reusser@usm.cl

[*] Correspondence: joelperez@uach.cl; Tel.: +56-9-79791242

[†] These authors contributed equally to this work.

**Abstract:** Nowadays, marine propulsion systems based on thermal machines that operate under the diesel cycle have positioned themselves as one of the main options for this type of applications. The main comparative advantages of diesel engines, compared to other propulsion systems based on thermal cycle engines, are the low specific fuel consumption of residual fuels, and their higher thermal efficiency. However, its main disadvantage lies in the emissions produced by the combustion of the residual fuels, such as carbon dioxide ($CO_2$), sulfur oxide ($SO_x$), and nitrogen oxide ($NO_x$). These emissions are directly related to the operating conditions of the propulsion system. Over the last decade, the International Maritime Organization (IMO) has adopted a series of regulations to reduce $CO_2$ emissions based on the introduction of an Energy Efficiency Design Index (EEDI) and an Energy Efficiency Operational Indicator (EEOI). In this context, adding a Shaft Generator (SG) to the propulsion system favoring lower EEDI and EEOI values. The present work proposes a selective control system and optimization scheme that allows operating the shaft generator in Power Take Off (PTO) or Power Take In (PTI) mode, ensuring that the main engine operates, always, at the optimum fuel efficiency point, thus ensuring minimum $CO_2$ emissions.

**Keywords:** marine propulsion system; shaft generator; power take-in; power take-off; energy efficiency design index; energy efficiency operational indicator; gradient vector optimization; power converter; torque oriented control

## 1. Introduction

Even though marine propulsion systems have been in constant development since the 18th century, nowadays the most common system used on board large carriers, i.e., container ships and tankers, is a system considering a diesel engine as the prime mover. Most of these engines are of the crosshead type, operating on the two-stroke cycle at low speed having long strokes, turbocharged, and directly coupled to a single fixed-pitch propeller. The power installed for these configurations vary from 10 MW and up to 80 MW [1]. These type of propulsion systems include the use of a Waste Heat Recovery System (WHRS), which relates taking the remaining heat of the exhaust gases generated from the combustion process of the diesel engine. This system was previously known as an economizer and was used to generate steam for heating processes only. Recently, in some cases, the WHRS uses heat to generate steam to be used in turbo-generators [2–4].

Diesel engines for marine applications have many advantages when compared to other prime movers such as turbines. They have a higher thermal efficiency and a low fuel oil consumption of

low-cost residual fuels. The disadvantage of consuming residual fuels is the high amount of $CO_2$, $SO_x$, and $NO_x$ emissions, which are related to the ship's operational condition [5,6]. Alternative fuels with low carbon and sulfur content have been considered to be used to replace these residuals fuels; however, its use in large carriers is not cost efficient and still presents a poor environmental performance. In this regard, alternative fuels such as Liquefied Natural Gas (LNG), biofuels and hydrogen are some of the most promising alternatives in study as replacement, but still some concerns about their storage, technological maturity and safety mitigating measures [7,8].

The ship's power design requirement represents the main constrain when the operational condition of the ship is assessed and compared to the power demands at normal operating conditions. The prime mover is forced, most of the time, to operate under underrated conditions increasing its fuel oil consumption therefore the amount of emissions. Diesel engines, as the one described, found their optimum fuel oil consumption point at ~75–80% of the MCR [9].

High fuel oil consumptions lead to an increase of emissions, e.g., $CO_2$ emissions, which have been monitored and since 2011 have been measured using an index called EEDI. This index is part of a mandatory regulation coming from the IMO and applicable to every new ship since 2013 [10,11]. An operational indicator called EEOI has also been considered but this is not mandatory yet, although is integral part of the Ship Energy Efficiency Monitoring Plant (SEEMP), which is mandatory to be implemented on board ships but that is not auditable by any means yet. The EEOI is used as a measuring tool to voluntary assess the efficiency of an existing ship.

The EEDI encourage the use of technologies such as shaft generator to reduce the use of the power installed on board through auxiliary generator sets [12]. This reduction of the use reduces the fuel oil consumption, therefore, as was mentioned before, reduces the amount of $CO_2$ emissions. The shaft generator force to use the diesel engine in a loading range, quite close to the optimum fuel oil consumption point [13,14]. The use of shaft generators above this point has been considering unjustified because of the possibility to overload the diesel engine leading to increase the fuel oil consumption.

The present work presents a marine diesel engine propulsion system with a direct driving shaft generator and a back to back converter based on the use of a selective control scheme. This scheme enables for the diesel engine to operate at its optimum fuel oil consumption point, which has been renamed as its Minimum Emissions Operating Point (MEOP). The scheme considers the use of the shaft generator as a Power Take Off (PTO) drive when the diesel engine operates below the MEOP and as a Power Take In (PTI) when the diesel engine operates above the MEOP. The shaft generator, at PTO, generates enough power to turn-off the generator set of the ship. These operational conditions have a repercussion on the EEOI, which is to be estimated and analyzed to prove the positive influence to lower the amount of emissions based on the reduction of the specific fuel oil consumption of the diesel engine. After EEOI results, the selective control scheme is going to be used to evaluate its influence over the EEDI of a new design looking for the development of an efficient propulsion system that ensures the compliance with the IMO regulations.

## 2. Background

Efficiency can be defined as the ratio of the useful work performed by a vessel to the total energy expended, but also can be expressed as actions designed to achieve efficiency [15]. Under these definitions, first, we can consider the vessel efficiency as the amount of fuel consumed, as the energy source to be expended, over the transport work performed by the vessel, as the process of carrying cargo, and second, we can consider vessel efficiency as the implementation of technological and operational means to a vessel to achieve higher levels of efficiency. Both definitions can be applied but for this research, the first is going to be applied over the design of new vessels, and the second over existing vessels. Both definitions can be worked together when analyzing the efficiency of a vessel, as to improve its efficiency in the design stage, it is required to know its current operational efficiency, from the EEOI. The current efficiency can be estimated when evaluating the amount of fuel that is consumed by the diesel engines of the main propulsion system and the auxiliary systems

when navigating. This estimation of vessel efficiency provides the baseline from where it is possible to improve it when considering the implementation of technological means, i.e., shaft generators and operational means, i.e., selective control schemes as the described in this research. The amount of fuel saved by the implementation of technological and operational means translated as the improvement into the efficiency of a vessel. Because the consumption of fuel generates emissions, any reduction of fuel consumption leads to lower emission levels. The use of the fuel consumption a as state variable provides the baseline of considering the use of EEDI and EEOI as means of evaluation of vessel design and operational efficiency.

### 2.1. Emissions from the Combustion Process

Emissions are generated during the process of converting the chemical energy of the fuel into mechanical work, Equation (1) represents the stoichiometric reaction of the fuel and the consequence emissions generation, $CO_2$ emissions are the higher amount of all of them.

$$C_mH_n + \left(m + \frac{n}{4}\right)O_2 + pN_2 \rightarrow m\,CO_2 + \frac{n}{2}H_2O + pN_2 \tag{1}$$

### 2.2. Fuel Oil Consumption and Diesel Engine and Shaft Generator Operation

The stoichiometric air to fuel ratio ($AFR_{st}$) is the minimum amount of air required to burn a kilogram of fuel and, when compared to the actual Air to Fuel Ratio (AFR), the stoichiometric ratio $\lambda$, presented in Equation (2), can be found [15]. The AFR can be considered at any engine load for the purposes of analysis.

$$\lambda = \frac{AFR}{AFR_{st}} \tag{2}$$

The engine's output power suffers when operating at lower loads condition, because at this condition, less fuel is available and the engine while trying to achieve a higher load demands more fuel to be provided to overcome the demand. The engine trying to maintain the power output at the desired operational condition increases the specific fuel oil consumption until it reaches the desired engine load, which is related to its speed as can be seen in Figure 1. At low engine load $\lambda \approx 4.0$, which decreases as the engine load is continuously increased. When the load is within the range of 75 to 80% of the engine maximum continuous rating (MCR), the value of $\lambda$ reaches its minimum. When going above 75–80% load, because higher amounts of air and fuel are required, $\lambda$ increases reaching a value of $\approx 2.0$ at the 100% of the MCR.

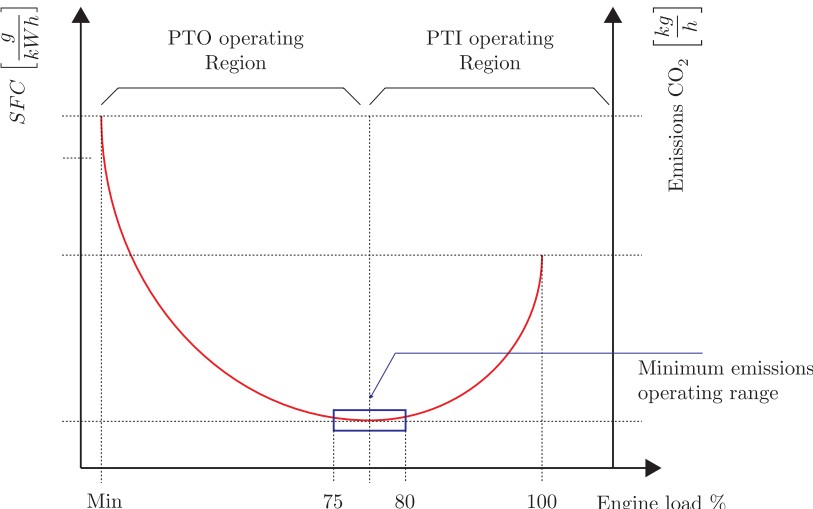

**Figure 1.** Power Take Off (PTO)/Power Take In (PTI) operating regions.

Figure 1 shows these $\lambda$ conditions and the optimal fuel oil consumption point or MEOP. The MEOP has been considered in this form because is a representation of the Specific Fuel oil Consumption (SFC), which is one of the factors to evaluate the amount of emissions generated by the engine when using the EEDI and EEOI.

Operationally, Figure 1 differentiate the operating regions of the shaft generator when considering the MEOP over the entire engine load range. Before MEOP the shaft generator operates as PTO and after MEOP operates as PTI. The shaft generator as PTO generates electricity to support the operation of the vessel and reduces the use of the diesel engines of the auxiliary system known as generation set. When operating as PTI, the electrical power to operate the shaft generator as an electric motor is provided by the generator set. Nonetheless, from this assumption, future work will investigate options to improve the efficiency of the generator set operation and the use of electric power sources, i.e., use of batteries and Non-Conventional Renewable Energies (NCRE) sources.

The engine efficiency at the MEOP is the highest and has been used as the evaluation point of the EEDI mandated by the IMO for every new ship constructed. The EEDI started with a minimum value established by 2013 and reduced by a percentage over the next 12 years [10,11], a low EEDI value means a more efficient ship, in terms of its design (hydrodynamics, propulsion system, and auxiliaries).

### 2.3. Energy Efficiency Design Index EEDI

The EEDI can be defined as a technical measure of $CO_2$ emissions per ship's capacity per nautical mile applied to new ship designs [10]. The EEDI equation is presented in Appendix A. Here, a modified version to be applied for the purposes of this paper is presented in Equation (3). This modified version accounts only for the main engine influence of emissions generation; therefore, it is an approximation that is going to be modified to establish and represent the influence of the shaft generator over the entire main and auxiliary systems of the ship. The auxiliary engines, shaft generator, and WHRS influence over the EEDI equation has been found minimum when compared to the main engine installed power therefore the EEDI value do not get really affected by them as stated in [11]. The influence of these factors is related to the operation of the ship.

$$EEDI = \frac{P_B \, SFC \, C_F}{DWT \, V_S} \tag{3}$$

The $P_B$ corresponds to the 75% rated installed brake power in kW, $C_F$ is the carbon factor in $g$ $CO_2$ per $g\,fuel$, DWT is the capacity of the ship in tonnes, and Vs is the ship's design speed in knots.

### 2.4. Energy Efficiency Operational Indicator EEOI

The MEOP has been also used to calculate the EEOI. The EEOI is the monitoring tool supporting the Ship Energy Efficiency Management Plan (SEEMP) applied to new and existing ships to measure the amount, in grams, of $CO_2$ per tonne cargo transported per nautical mile for a single voyage [11]. The equation to calculate EEOI is presented in Appendix B and here a modified version to be applied for the purposes of this paper is presented in Equation (4). This modified version uses the SFC instead of the total amount of fuel consumed for a single voyage to relate the indicator with the operational performance described to evaluate EEDI and have a comparison point of the ship's design performance and the actual operation at the required MEOP at low and high loads. The $m_c$ factor accounts for the mass of cargo transported in tonnes and $D$ to the distance, in nautical miles, of the cargo transported.

$$EEOI = \frac{SFC \, C_F}{m_C \, D} \tag{4}$$

Having a comparison point between the design and the operational behavior of the ship allows for a better understanding of these tools to evaluate the current efficiency of the ship, also allowing to consider technological and operational options to improve ship's efficiency. The use of a shaft

generator is part of these improvements, and its influence into of ship's efficiency is presented when analysing the Shaft Generator/Motors Emissions Factor $f_{gef}$ into the EEDI equation, this factor is presented in Equation (5), where the power generated accounts and is related to the power generated by the auxiliary engines or generator set to support the service of the ship.

$$f_{gef} = \left( f_i\, P_{PTI} - f_{eff}\, P_A E \right) C_{FAE}\, SFC_{AE} \tag{5}$$

Equation (5) can also be contrasted with the Efficiency Technology Factor (ETF) presented in Equation (6), which relates the reduction in power requirements that any technology generates and is used to improve the ship's efficiency.

$$ETF = f_{eff}\, P_{eff}\, C_F\, SFC \tag{6}$$

The efficiency technology factor $f_{eff}$ in Equations (5) and (6) represents the percentage of influence of the power output of the technology and relates to its efficiency. The background presented allows for the introduction of the control scheme selected as the most appropriate to represent the influence of a shaft generator into the propulsion system of a ship, because relates its fuel consumption and its emissions in accordance with known and validated indexes for ship's efficiency evaluation.

## 3. Hybrid Propulsion System Characterization

The hybrid propulsion system under study consists of low speed 2-stroke diesel engine, driving directly the propulsion shaft. A permanent magnet synchronous machine is coupled to the diesel engine using a single-stage gearbox. This geared mechanical transmission system enables the combination of the mechanical and electrical prime movers in the same kinematic drivetrain, as shown in Figure 2

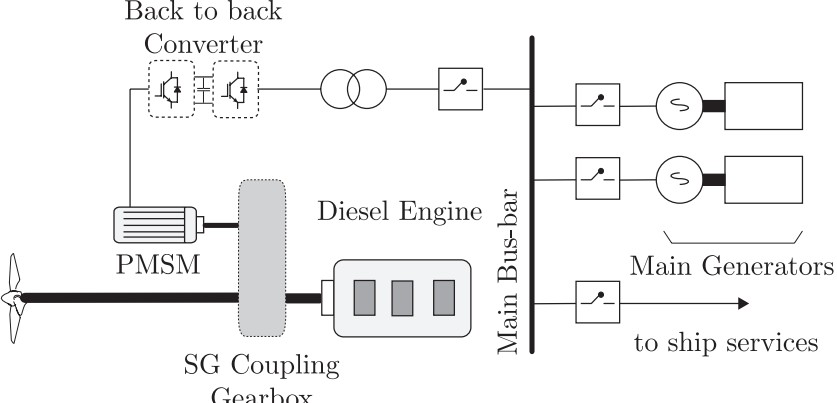

**Figure 2.** Hybrid propulsion system configuration.

The synchronous machine is connected to the main ship's grid using a back-to-back power converter. This configuration enables bidirectional power flow, between the electric drive and the ship's grid, thus enabling the electric drive to operate in power take-off (PTO) or power take-in (PTI) modes, depending on the direction of the power flow, as shown in Figure 3. Moreover, the use of the back-to-back converter configuration decouples the electric drive and grid control schemes. This enables the grid side control scheme to be synchronized referred to the ship's main busbar frequency, independently of the electric drive operational frequency, and therefore the diesel engine operational speed [16].

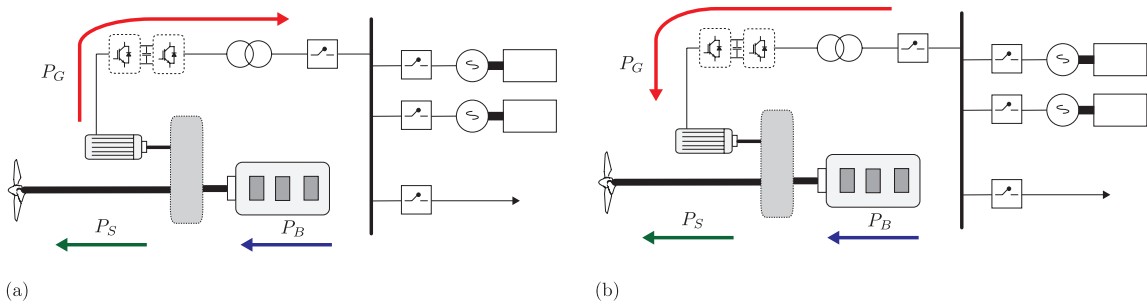

**Figure 3.** Power flow in (**a**) power take-off operation mode; (**b**) power take-in operation mode.

**Power take-off mode:** in this mode, the main diesel engine supplies the power needed for the propulsion $P_S$ as well as for the ship's consumers $P_G$ by forcing the electric drive to operate within the generator region. Depending on the ship's load and the required propulsion power, all or some of the generator sets (GS) are turned off, as shown in Figure 3a. The engine developed power $P_B = P_S + P_G$.

**Power take in mode:** in this mode the electric drive is forced to operate within the motor region, an auxiliary motor, allowing to reduce the main engine's load, as shown in Figure 3b. Depending on the required shaft power, the system operates in booster mode, when both electric and diesel provide deliver power to shaft, or diesel-electric mode, when only the electric drive delivers power. Therefore, the engine developed power $P_B = P_S - P_G$.

### 3.1. Diesel Engine Model

The diesel engine dynamics is obtained energy conversion principle, by defining the system's Hamiltonian $H(x)$ as given in Equations (7)–(9)

$$H(x) = W_i - \sum_{\ell=1}^{n} Ec_\ell + W_I \tag{7}$$

where $W_i$ is the chemical energy also known as indicated energy, $Ec$ correspond to the system loses, and $W_I$ is the stored energy in the inertia. The indicated energy can be defined as a nonlinear function of the fuel enthalpy $h$, the fuel flow rate $g$, and shaft rotational speed $\dot{\theta}_r$

$$W_i = f_c\left(h, g, \dot{\theta}_r\right) \tag{8}$$

and the energy stored in the inertia is given as in Equation (9)

$$W_I = \frac{1}{2} L \frac{d}{dt} \theta_r \tag{9}$$

where $L$ stands for the rotational momentum. The total converted energy into mechanical torque is given by Equations (10) and (11).

$$\frac{\partial}{\partial \theta_r} H(x) = 0 \tag{10}$$

$$J \frac{d^2}{dt^2} \theta_r = T_i - T_p - T_f \tag{11}$$

where $T_i$ corresponds to engine's indicated torque, $T_p$ the pumping torque, and $T_f$ the friction torque.

The indicated torque is dependent on the amount of fuel injected into each of cylinders per cycle as given in Equation (12)

$$T_i = \frac{m_{cy} \, n_{cy} \, \rho_h \, \eta_{ig}}{2 \, \pi \, n_{cs}} \tag{12}$$

where $m_{cy}$ corresponds to the fuel delivery per cycle per cylinder, $n_{cy}$ to number of cylinders, $\rho_h$ is the heating value of fuel, $\eta_{ig}$ is the indicated efficiency, and $n_{cs}$ the number of crank revolutions. The indicated efficiency is given as in Equation (13)

$$\eta_{ig} = \eta_{cc} \left( 1 - \frac{1}{r_c^{\gamma-1}} \right) \tag{13}$$

where $r_c$ is the compression ratio, $\gamma$ the gas specific heat capacity ratio in the cylinder, and $\eta_{cc}$ represents the combustion chamber efficiency.

As presented, the indicated torque is highly dependent on several engine parameters, such as the number of cylinders, the fuel delivery per cycle, the compression ratio, and combustion chamber efficiency. On the other hand, the diesel engine emissions are defined by its residual gas fraction $\chi_r$, which represents a measure of $CO_2$ concentrations of the working gas in the compression stroke, during the energy conversion process, as defined in Equation (14)

$$\chi_r = \frac{(\tilde{\chi}_{co_2})_C}{(\tilde{\chi}_{co_2})_E} \tag{14}$$

where $(\tilde{\chi}_{co_2})_C$ and $(\tilde{\chi}_{co_2})_E$ stands for the $CO_2$ fractions during compression and exhaust, respectively.

The torque component corresponding to energy losses $T_C$ is given by Equations (15) and (16)

$$T_C = \frac{\partial}{\partial \theta_r} \sum_{\ell=1}^{n} Ec_\ell \tag{15}$$

$$T_C = T_p + T_f \tag{16}$$

where the pumping torque $T_p$ and friction torque $T_f$ can be expressed as in Equations (17) and (18), respectively.

$$T_p = \frac{V_d}{2\,\pi\,n_{cs}} \left( p_{em} - p_{im} \right) \tag{17}$$

$$T_f = \frac{V_d}{2\,\pi\,n_{cs}} \left( c_0 + c_1\,n_r + c_2\,n_r^2 \right) \tag{18}$$

here $V_d$ corresponds to the engine displacement volume, $p_{em}$ is the exhaust pressure to the manifold, and $p_{im}$ the manifold inlet pressure. The friction torque $T_f$, on the other hand, may be assumed to be a quadratic polynomial depended on the engine revolutions $n_r$, with $c_0$, $c_1$, and $c_2$ fitting constants.

From Equations (7)–(17) it becomes self-evident that the diesel engine mathematical model is highly nonlinear and dependent on several specific construction parameters. However, a linearized model may be used, considering all important nonlinear characteristics [17–20], which can be modeled as dead-times and time-delays contained in $\tau_1$ and $\tau_2$, respectively, and constant parameters $k_1$, $k_2$, as presented in Equations (19)–(20)

$$\frac{d}{dt}\,y = -\frac{1}{\tau_1}\,y + \frac{k_1}{\tau_1}\,u \tag{19}$$

$$J\frac{d}{dt}\,\omega_r = -B\,\omega_r + k_2\,y\,(t - \tau_2) - T_L \tag{20}$$

where $J$ stands for the engine's inertia, $B$ is the friction coefficient, $\omega_r$ corresponds to the engine rotational speed, $u$ to the speed controller output, $y$ the position of the fuel rail, and $T_L$ the external load torque. Values for $k_1$, $k_2$, $\tau_1$, and $\tau_2$ may be found empirically or from the data provided by the manufacturer using a model fitting algorithm, as stated in [21].

Considering the previously made considerations and modeling restrictions, it is possible to build an emissions model, on the basis of the data provided by the manufacturer, using an appropriate polynomial approximation with squares regression.

Given $m$ data points $\{x_i\,y_i\}_{i=1}^m$ with $x_i$ given output power and $y_i$ corresponding $CO_2$ emissions rate; the best fit polynomial for the $CO_2$ emissions $e(x)$ could be developed using Equation (21)

$$e(x) = \sum_{k=0}^{n} \alpha_k\, x^k \quad n < m-1 \tag{21}$$

where $\alpha_k\ \forall\, k$ coefficients may be found by minimizing the least square error using Equation (22)

$$A^T A\, \boldsymbol{a} = A^T \boldsymbol{y} \tag{22}$$

with the coefficients vector $\boldsymbol{a} = [\alpha_0 \ldots \alpha_n]^T$, the sample value vector $\boldsymbol{y} = [y_0 \ldots y_n]^T$, and $A$ the Vandermonde matrix, given as in Equations (23) and (24)

$$A = \begin{bmatrix} 1 & x_1 & x_1^2 & \ldots & x_1^n \\ 1 & x_2 & x_2^2 & \ldots & x_2^n \\ \vdots & \vdots & \vdots & & \vdots \\ 1 & x_m & x_m^2 & \ldots & x_m^n \end{bmatrix} \quad \forall x_i\ i = 1, \ldots, m \tag{23}$$

$$\boldsymbol{a} = (A^T A)^{-1} A^T \boldsymbol{y} \tag{24}$$

obtaining finally an $n$ degree polynomial representing the $CO_2$ emissions profile, as given in Equation (25)

$$y(x) = a_0 + a_1\, x + a_2\, x^2 + \ldots + a_n\, x^n \tag{25}$$

### 3.2. Electric Drive Model and Control

The anisotropic permanent magnet synchronous machine (PMSM) mathematical model in an arbitrary synchronous reference frame $d\,q$ is described as in Equations (26) and (27) [22],

$$v_s^{(dq)} = R_s\, i_s^{(dq)} + \frac{d}{dt}\psi_s^{(dq)} + \boldsymbol{F}\,\psi_s^{(dq)} \tag{26}$$

$$\boldsymbol{F} = \begin{bmatrix} 0 & -\omega_k \\ \omega_k & 0 \end{bmatrix} \tag{27}$$

where $v_s$ corresponds to the stator voltage, $R_s$ to the stator resistance, $i_s$ the stator current, $\psi_s$ the stator flux linkages, and $\omega_k$ to the shaft synchronous speed. The stator flux linkages are given as in Equations (28) and (29):

$$\psi_s^{(dq)} = \boldsymbol{G}\, i_s^{(dq)} + \boldsymbol{P}\psi_m \tag{28}$$

$$\boldsymbol{G} = \begin{bmatrix} L_d & 0 \\ 0 & L_q \end{bmatrix} \quad \boldsymbol{P} = \begin{bmatrix} 1 \\ 0 \end{bmatrix} \tag{29}$$

$L_d$ and $L_q$ are the direct and quadrature reference frame inductances, respectively, and $\psi_m$ the permanent magnet flux linkage. The electromechanical torque developed by the PMSM is given in Equations (30) and (31)

$$T_e = \frac{\partial}{\partial\,\theta_r} W_{fld}\left(\psi_s^{(dq)},\, i_s^{(dq)},\, \theta_r\right) \tag{30}$$

$$T_e = \frac{3}{2}\, p\left\{\psi_m\, i_s^q + (L_d - L_q)\, i_s^d\, i_s^q\right\} \tag{31}$$

where $p$ are the number of pole pairs.

Despite the classical FOC control scheme, which is used to control the drive shaft speed, in this case, the control objective is to control the torque developed by the electric drive [23], which is

achieved by means of the electric torque reference, provided by the optimization algorithm output. The implementation of the electric drive control scheme is shown in Figure 4.

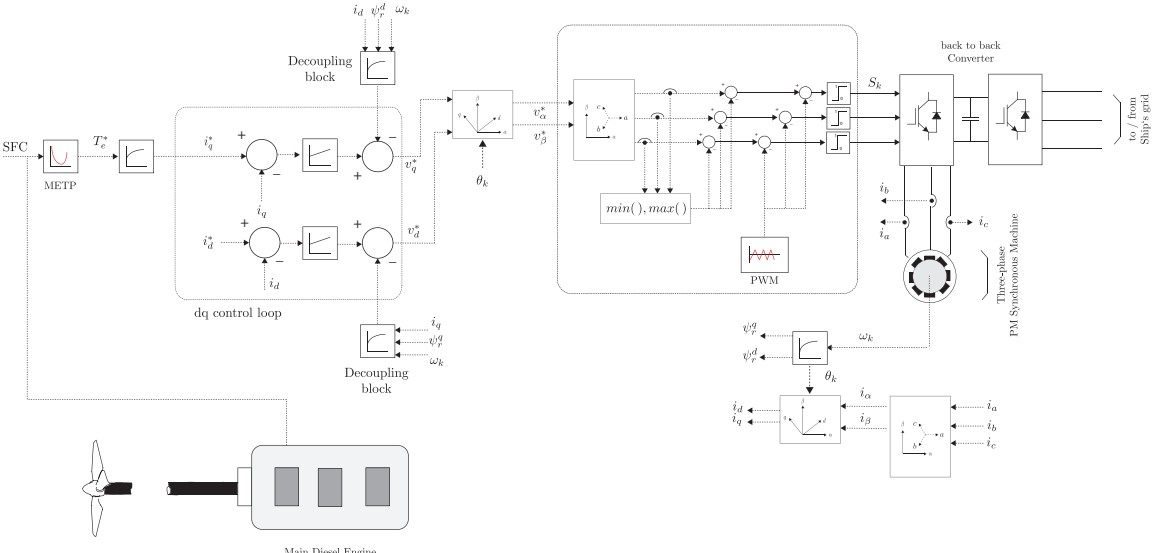

**Figure 4.** Torque field-oriented control scheme.

### 3.3. Grid Side Power Flow Control

Grid side control is achieved by implementing active and reactive power control [24,25], using a virtual flux voltage oriented control (VF-VOC) strategy. Active and reactive power, $P$ and $Q$, respectively, in a synchronous rotating reference frame, grid side-oriented $u\,v$ are given in Equations (32) and (33), as a result of using a voltage orientation in $u$ coordinate.

$$P = \frac{3}{2} \mathbb{R}e \left\{ v^u \left( i^u + j i^v \right) \right\} \tag{32}$$

$$Q = \frac{3}{2} \mathbb{I}m \left\{ v^u \left( i^u + j i^v \right) \right\} \tag{33}$$

Thus, by setting the reactive component of the grid current $i^v = 0$ it is possible to maximize the active power flow into the grid. Orientation into the grid side synchronous reference frame $u\,v$ is achieved by extracting the orientation angle $\theta_p$ provided by a virtual-flux space vector $\psi^{(xy)}$ referred to the voltage drop in the output inductance $v_o^{(xy)}$ as in Equations (34) and (35). Implementation of the grid side control scheme is provided in Figure 5.

$$\psi^{(xy)} = \int v_o^{(xy)} (t)\, dt \tag{34}$$

$$\theta_p = \mathrm{atan2} \left( \psi^x,\ \psi^y \right) \tag{35}$$

The corresponding grid side dynamic model in the $u\,v$ synchronous reference frame, is given in Equation (36),

$$v^{(uv)} = R\, i^{(uv)} + L\, \frac{d}{dt} i^{(uv)} + \boldsymbol{F}\, L\, i^{(uv)} + v_g^{(uv)} \tag{36}$$

where $v^{(uv)}$ corresponds to the converter output voltage, $i^{(uv)}$ stands for the grid side current, and $v_g^{(uv)}$ to the main busbar voltage, in the $u\,v$ reference frame. Line parameters of resistance and inductance are given as $R$ and $L$, respectively, and matrix $\boldsymbol{F}$ has been defined in Equation (27).

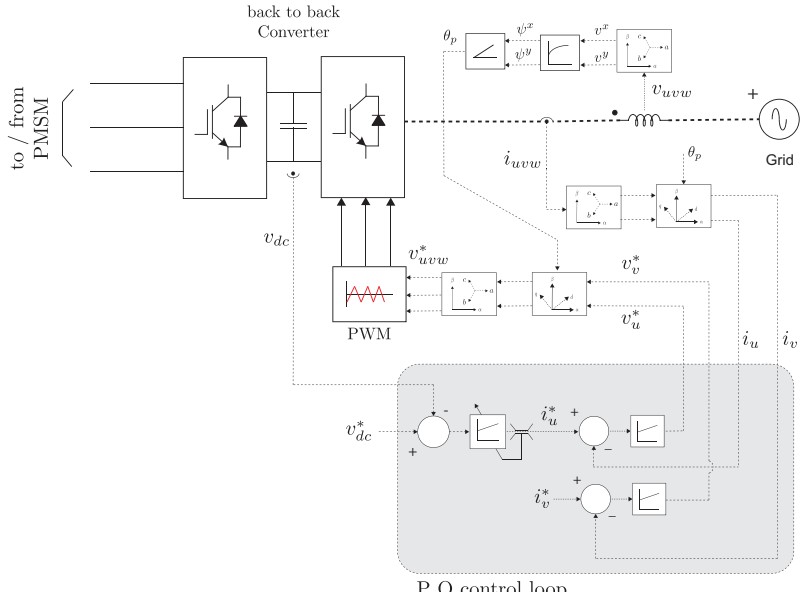

**Figure 5.** Voltage-oriented control scheme.

## 4. Optimization Strategy

Let us define an arbitrary optimization problem $\phi$ as in Equations (37)–(40), given a set of candidate solutions $\mathcal{C}$, a set of solution $\mathcal{S} \subseteq \mathcal{C}$, an objective function $f(x)$, and $\nu$ the optimization sense.

$$\phi = \langle\, \mathcal{C}\,,\, \mathcal{S}\,,\, \nu\,,\, f(x)\,\rangle \tag{37}$$

$$\mathcal{C} = x \quad;\ x = \{\, x_o\,,\, \dots\,,\, x_n\,\} \tag{38}$$

$$\mathcal{S} = x \pm \delta \tag{39}$$

$$\nu = min\,\{\, f(x)\,|_{x\pm\delta}\,\} \tag{40}$$

where $x$ corresponds to the system state and $\delta$ to the variation of the state introduced by the search direction of the optimization strategy.

The implemented optimization strategy is based on the use of the gradient vector $\nabla f(x)$ as search direction for each iteration. Note that the gradient vector is orthogonal to the plane tangent to the contour surfaces of the function to optimize; $\nabla f(x) = g(x) = [\, \frac{\partial f}{\partial x_1}\ \dots\ \frac{\partial f}{\partial x_n}\,]^T$. The gradient vector at a point $g(x_k)$ represents the direction of maximum rate of change, which is given by $|\,g(x_k)\,|$

The optimization strategy searches for the point $x_k$, where $|\,g(x_k)\,| \leq \mu_g$, given an initial state $x_o$; certain convergence parameters $\mu_g$, $\mu_a$, and $\mu_g$; and a normalized search direction $p_k$; given Equations (41) and (43)

$$p_k = -\frac{g(x_k)}{|\,g(x_k)\,|} \tag{41}$$

$$x_{k+1} = x_k + \alpha\, x_k \tag{42}$$

for some $\alpha$ such that satisfies Equation (43)

$$|\,f(x_{k+1}) - f(x_k)\,| \leq \mu_a + \mu_r\,|\,f(x_k)\,| \tag{43}$$

The search objective corresponds to the minimum $CO_2$ emissions operating point of the diesel engine, and the function to optimize $f(x)$, to the diesel engine emissions profile, given a a certain required output power. The result from the optimization problem is used as torque reference for the TFOC electric drive control scheme. Figure 6 shows the algorithm implementation.

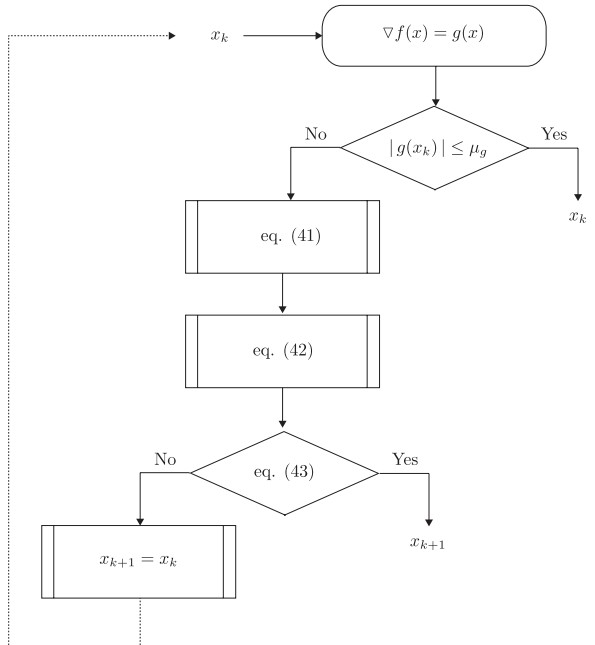

**Figure 6.** Gradient-based optimization algorithm structure.

## 5. Simulation Results

In this section, the performance of the diesel engine and the the optimization scheme under different load and operational conditions are presented. The hybrid propulsion system model under study was developed using PLECS. The controllers for the electric drive and the grid side, as well as the optimization algorithm, were developed and implemented in C code.

### 5.1. Diesel Engine Performance

Results are presented differentiating the performance of the control scheme of the shaft generator and the results of the fuel consumed by the engine when simulating the PTO and PTI conditions. The methodology considers the use of the IMOs EEDI and EEOI tools to show the benefits of the scheme, providing a reference to evaluate the design efficiency and the operational efficiency.

Results are plotted over the two simulation conditions considered but presented over a power range to simulate specific operational conditions such as slow steaming. This operational condition has been considered because represents the ability of the simulation to show the performance of the ship at low ship's speed, which can be used to evaluate a ship design over a higher range of options to get an efficient design. A more accurate evaluation of the EEDI could be necessary but still results are providing a great assertiveness of the methodology selected.

The data used to simulate the performance of the control scheme considers the use of a ship, which has specific information: its capacity, speed, cargo transported, distance navigated, power installed, and the type of fuel consumed. The type of ship considered was selected from a worldwide database of ships [26]. When analyzing the database and the specific information needed to evaluate the design and operational efficiencies of a ship, very large crude oil carriers were the type of ships more reluctant to be used because of the simplicity of their propulsion system and the significance of the amount and type of fuel consumed. The propulsion system consists of a diesel engine directly coupled to a fixed pitch propeller. The auxiliary power installed for this specific type of ship accounts for ~10% of the propulsion power installed [26]. Table 1 presents the open source data used for simulation that were fixed as the initial conditions. The range of data is only a reference of the type of ship found in the database and no consideration to the operational profile of them has been considered for simulation purposes.

**Table 1.** Operational parameters for simulation.

| Type of Ship | Fuel Consumed | Capacity (dwt) | Speed (kn) | Cargo Transported (tonnes) | Installed Power (MW) |
|---|---|---|---|---|---|
| Very Large Crude Oil Carrier (VLCC) | Heavy Fuel Oil (HFO) | 300,000 | 15 | 315,000 | 25 |
| | | 320,000 | 21 | 330,000 | 36 |

The engine selected to be modeled and evaluated is an engine from MAN [27]. The 7G80ME-C9.2-TII diesel engine was selected having a specified maximum continuous rating (SMCR) power of 33 MW at 72 rpms. The specific fuel oil consumptions (SFC) vary from 187.1 g/kWh at low engine load to 166 g/kWh at high engine load. The normal continuous rating of the engine was considered ~75% of the SMCR. The total power delivered by the auxiliary generators during the navigation has been considered ≈3% of the propulsion engine rated power. When consuming HFO, a carbon factor of 3.114 g $CO_2$/g Fuel was used to estimate the emissions. With this information, the simulation of the control scheme was carried out and the results are presented next.

*5.2. Control Scheme Performance at PTO Operating Region*

Results are presented considering the performance of the shaft generator operating as a PTO, including the power and fuel consumption performance and the evaluation of the EEDI and EEOI tools.

The control scheme was evaluated considering a navigation time period long enough to simulate a consistent increase of the brake power of the engine over a step time to reflect its performance and SFC variation to get a steady evaluation of the efficiencies. Blue curves in Figure 7 shows the results of a segment of the PTO operating region while simulating a period of navigation time of 5 h, where, in the left-hand side of the figure, it is possible to appreciate how the brake power lineally increases from 5 MW to 12 MW. The right-hand side of the figure shows the decrease of the SFC from a maximum value at low engine load, the power delivered as PTO is stable enough to allow for the ship to turn-off the generator set. The red curves are the results of not having a shaft generator installed. The increase in the brake power when using a shaft generator is ~5% of the SMCR.

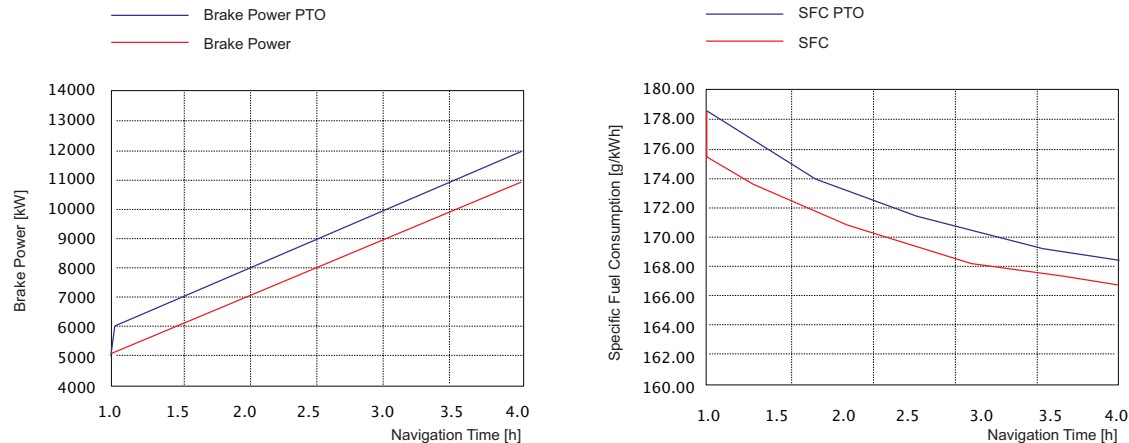

**Figure 7.** PTO evaluation performance.

When evaluating the EEDI, a specific value of 27,412 g $CO_2$/g Fuel was calculated. This value represents the amount of $CO_2$ emissions by design, which is therefore a value that can be modified at design stages only when the main and auxiliary machinery are selected and allocated to the vessel. A better approach could be to install a less powerful engine, but that means a completely different approach of the design spiral of the new ship. Following this evaluation, EEOI has been considered because represents the ability to calculate the emissions of the ship when in service navigating different

routes. EEOI allows to check the variations of the same parameters that EEDI uses to be evaluated, therefore provides with a more comprehensive form to understand the operational behavior of a well design ship.

EEDI provides a fixed value at the design conditions, yet EEOI can be used to evaluate the performance at every variation of engine load. EEOI considers the total amount of fuel consumed and mass of cargo transported and distance navigated, the latests being just another representation of cargo capacity and ship's speed, respectively.

One of the main objectives of this work was to evaluate, using EEDI and EEOI, the influence of a shaft generator when applying a control scheme of its operation. The purpose is looking for reduction into the SFC at different engine loads. Also, to prove that the reduction of the SFC compared to the increase into the necessary power to be developed, to overcome the extra necessary brake power to be produced, to propel the ship and to use the shaft generator as PTO and PTI, respectively.

Figure 8 shows the results of the SFC variation at PTO operating region when applying the proposed control scheme, red curve, and the blue curve shows the SFC variation when not having a shaft generator. The SFC differences are between 0.1% to 2% over the whole engine load range plotted, the difference even though can be considered small is quite significant when evaluating the EEOI, having the maximum SFC reduction between 6000 kW and 7000 kW. The difference is barely noticed because of the scale of the plotted results.

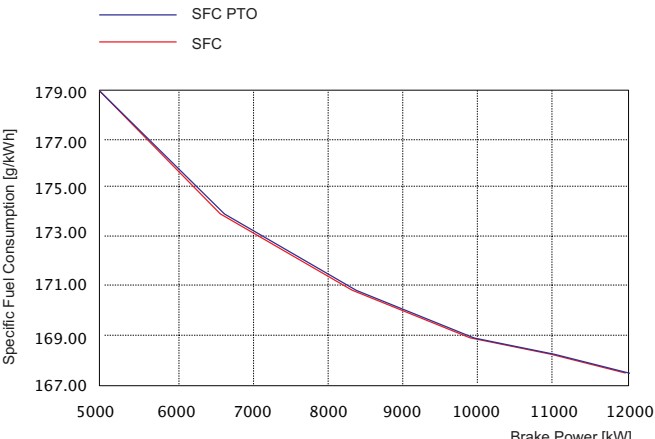

**Figure 8.** Specific Fuel oil Consumption (SFC) difference at PTO and without shaft generator.

The EEOI values are presented in Figure 9 together with the SFC obtained when using the shaft generator control scheme. The SFC decreases while the engine load increases. The fuel consumed provides EEOI values in accordance with its consumption over the navigation period and the navigated distance, as was described and established before as the initial conditions for simulation. The EEOI increases, but at a low rate over the engine load, which is a consequence of the decrease in the SFC. Although the decrease of the SFC is not quite significant allows for a low increment of the EEOI value, which leads to a overall reduction of the amount of operational $CO_2$ emissions.

Figure 10 shows the results of comparing the EEOI values of the applied control scheme, red curve, against not having a shaft generator installed, blue curve. Results are showing that the control scheme applied makes the ship to reduce the amount of operational $CO_2$ emissions even though an extra amount of brake power is necessary to be generated providing great assertiveness of the methodology and the SFC reduction over the period when operating at the PTO operating region.

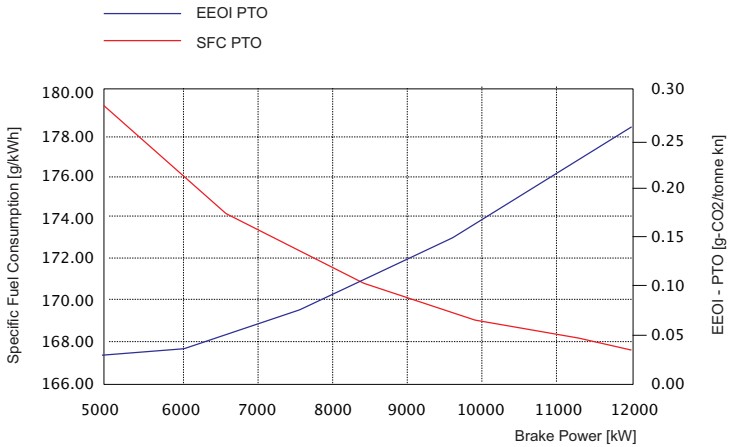

**Figure 9.** EEOI against SFC–PTO.

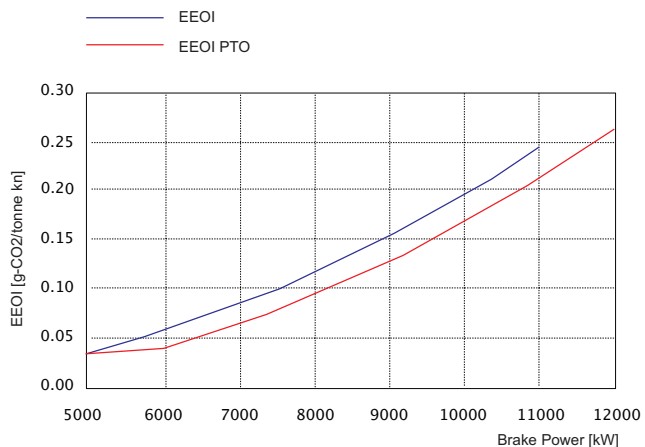

**Figure 10.** EEOI comparison.

### 5.3. Control Scheme Performance at PTI Operating Region

Results are presented considering the performance of the shaft generator operating as a PTI including the power and fuel consumption performance and the evaluation of the EEDI and EEOI tools. Following the same procedure to describe the results of the PTO operating region, the PTI operating region results are shown.

The blue curves in Figure 11 show the results of a segment of the PTI operating region while simulating a period of navigation time of 2.5 h, where, in the left hand side of the figure, it is possible to appreciate how the brake power lineally increases from 24 MW to 27 MW when not having a shaft generator, red curve, which also means an increase in the SFC as can be seen in the right-hand side of the figure. When applying the control scheme to operate the shaft generator, the PTI reduces the brake power, blue curve on the left-hand side of the figure therefore, a reduction of the SFC as can be seen in the right-hand side of the figure. The PTI reduces the brake power ~5% of the SMCR.

Figure 12 shows the results of the SFC variation at PTI operating region when applying the proposed control scheme, blue curve, and the red curve shows the SFC variation when not having a shaft generator.

The SFC differences are between −0.4% to 1.5% over the whole engine load range plotted. The difference reflects that even though the engine load increases, for PTI operation, the SFC decreases because the engine goes back to the high efficiency region operation or MEOP described before. On the other hand, the engine without a shaft generator increases the SFC, as expected, because of the operation away of the MEOP.

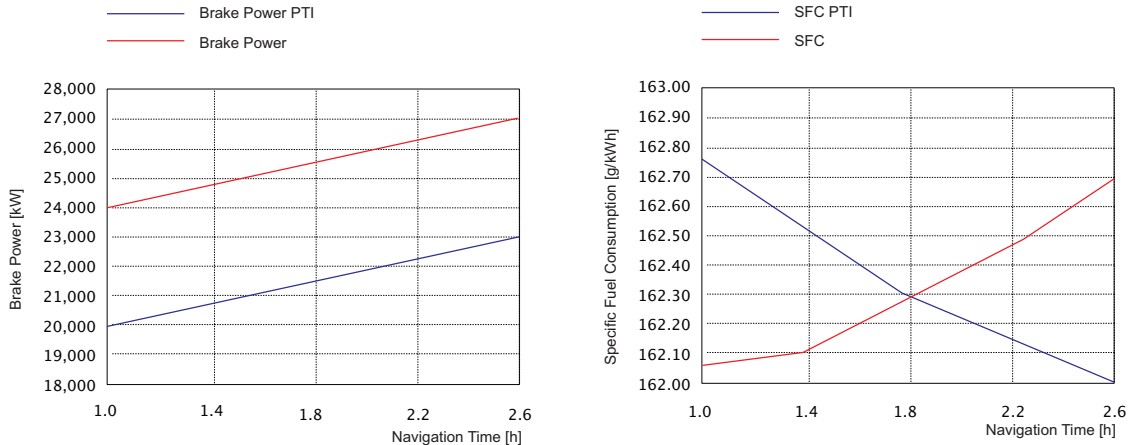

**Figure 11.** PTI evaluation performance.

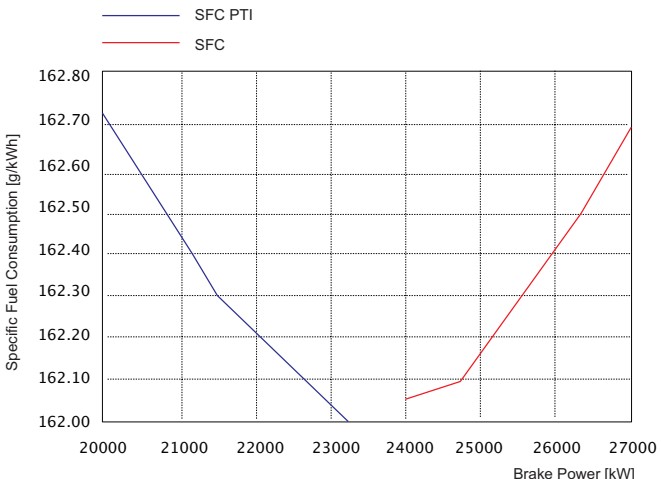

**Figure 12.** SFC difference at PTI and without shaft generator.

EEOI values are presented in Figure 13 against the SFC obtained when applying the shaft generator control scheme. The SFC decreases while the engine load increases, which gives a set of EEOI values in accordance with the amount of fuel consumed over the navigation period and the navigated distance. The lineal increment of the EEOI is considered low over the engine load and reflects the decrease of the SFC because of the control scheme applied. The operational emissions are reduced in accordance with the reduction of the SFC.

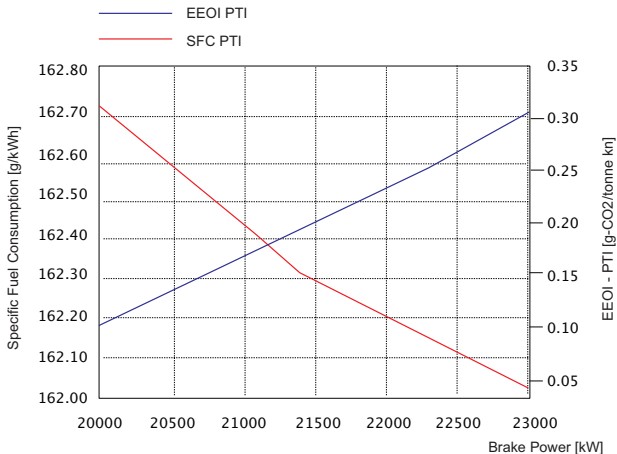

**Figure 13.** EEOI against SFC at PTI condition.

Figure 14 shows the results of comparing the EEOI values of the applied control scheme, blue curve, against not having a shaft generator installed, red curve.

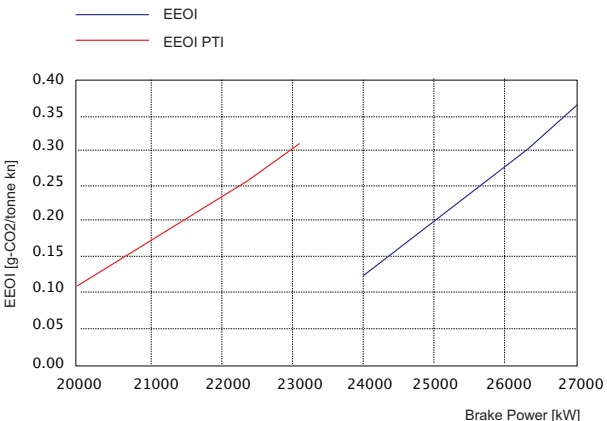

**Figure 14.** EEOI comparison.

Results are showing that the control scheme applied makes the ship to reduce the amount of operational $CO_2$ emissions because is making the engine to work closer to MEOP providing great assertiveness of the methodology.

*5.4. Electric Drive Performance*

In this section, an evaluation of the proposed control scheme, using a Minimum Emissions Point Tracking algorithm, based on a gradient vector optimization technique, is presented. Simulation results include two different torque steps at ① and at ②, both representing different shaft power operational conditions $P_s$ @ 90% rated power as base condition before ①, de-rating to 60% of rated power in ①, and finally going to 80% rated power at ②.

Figure 15 shows the performance of the electric drive (the performance of the PMSM) in terms of its controlled currents. At low operational load below 75% of rated power, the MEPT algorithm sets the electric machine's torque reference in the generator region, therefore the torque producing current $i_q < 0$, whereas the flux producing current $i_d = 0$. On the other hand, when entering into a high load condition above the minimum SFC point, the MEPT, forces the PMSM to operate in the motor region, therefore the torque producing current $i_q > 0$, while the flux producing current $i_d = 0$.

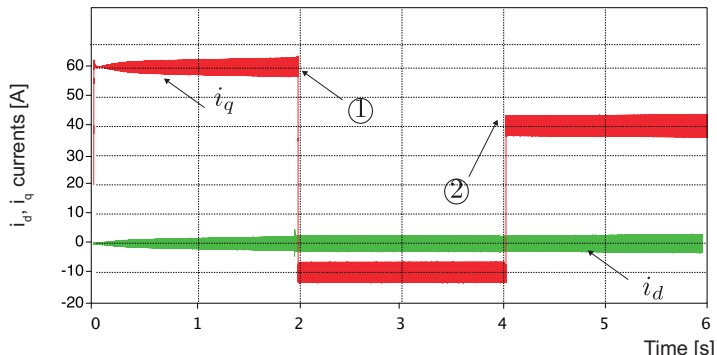

**Figure 15.** Electric drive performance $i_d$ $i_q$ during torque demand step.

In Figure 16 the tracking performance of the optimization algorithm is shown, in the presence of a step change of the diesel engine torque. As shown, the torque electric reference $T_e^*$ presents a fast and accurate response and keeps on tracking the minimum emissions operation point.

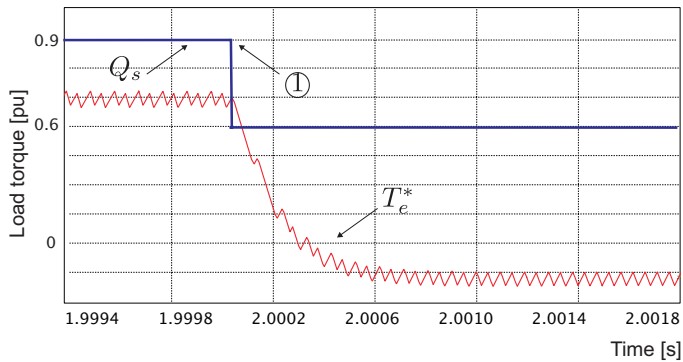

**Figure 16.** Optimization algorithm step dynamic response.

The grid side currents dynamic performance is shown in Figure 17, showing a sinusoidal behavior and fast dynamic response, during the transition from PTI to PTO mode at ① and from PTO to PTI operation at ②. Moreover, the fast tracking dynamics of the optimization algorithm, ensures minimization on the current wave-form distortion. On the other hand, due the fact that the grid side control scheme is decoupled from the electric drive control scheme, and ensures zero reactive power flow, the amplitude of the grid side currents are dependent on the electric drive's torque reference.

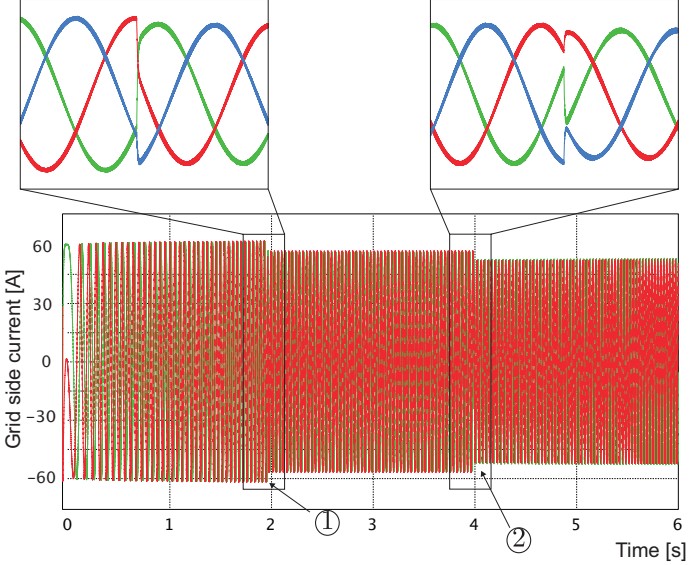

**Figure 17.** Grid side currents performance.

## 6. Conclusions and Future Work

Results of the fuel consumed by the engine when simulating the PTO and PTI conditions are not yet significant in helping to reduce EEOI values at both operating regions, which means less operational $CO_2$ emissions.

The control scheme, when simulating the engine performance at PTI operating region, makes the engine work closer to the MEOP, which leads to low SFC, and therefore low operational emissions.

The control scheme shows the high reliability and accuracy to follow the optimization algorithm with good dynamics, at both operating conditions as PTI and PTO. It also ensures bidirectional power flow, with low distortion of in the grid side currents.

The optimization function presents an accurate performance to obtain a local search for the minimum emissions point, starting at a random state. Future results may include the use of an adaptive perturbation function to ensure full convergence when reaching the minimum emissions point.

Slow steaming was mentioned because of the application of the scheme yet needs to be worked out separately from a design stage to provide more accurate conclusions to this work.

The ship and engine data considered for the simulations is open source and provides great value to continue to be used in this research.

Future work will consider the application of the proposed hybrid propulsion control scheme in a small-scale vessel for experimental validation of the SFC performance, as well as adding a more accurate operational profile of the vessel to work in depth the auxiliary engines and WHRS operational emissions generation influence.

**Author Contributions:** Conceptualization, J.R.P. and C.A.R.; methodology, J.R.P. and C.A.R.; simulation and programming, C.A.R.; validation, J.R.P. and C.A.R.; formal analysis, J.R.P.; investigation, J.R.P. and C.A.R.; data curation, J.R.P.; writing—original draft preparation, J.R.P. and C.A.R.; writing—review and editing, J.R.P. and C.A.R. All authors have read and agreed to the published version of the manuscript.

**Funding:** This research received no external funding.

**Acknowledgments:** The authors wish to thank the financial support from the Chilean Found for Human Resource Development (CONYCIT) through its Ph.D. scholarships (CONICYT/21130448).

**Conflicts of Interest:** The authors declare no conflicts of interest.

## Abbreviations

The following abbreviations are used in this manuscript:

| | |
|---|---|
| AFR | Air to fuel ratio |
| EEDI | Energy Efficiency Design Index |
| EEOI | Energy Efficiency Operational Indicators |
| FOC | Field oriented control |
| IMO | International Maritime Organization |
| LNG | Liquefied Natural Gas |
| MCR | Maximum continuous rating |
| MEOP | Minimum Emissions Operating Point |
| NCRE | Non-conventional renewable energies |
| PMSM | Permanent magnet synchronous machine |
| PTI | Power Take-In |
| PTO | Power Take-Off |
| SEEMP | Energy Efficiency Monitoring Plant |
| SFC | Specific fuel oil consumption |
| SG | Shaft Generator |
| SMCR | Specified maximum continuous rating |
| TFOC | Torque field oriented control |
| VF-VOC | Virtual flux voltage oriented control |
| WHRS | Waste Heat Recovery System |

## Appendix A

$$EEDI = \frac{A + B + C - D}{E} \tag{A1}$$

- Main Engines Emissions: Factor $A$

$$A = \left( \prod_{j=1}^{m} f_j \right) \left( \sum_{i=1}^{n} P_{ME(i)}\, c_{F\,ME(i)}\, SFC_{ME(i)} \right) \tag{A2}$$

| | |
|---|---|
| $f_i$ | Correction factor for ship specific design elements. |
| $P_{ME}$ | Power of main engines. |
| $C_{F\,ME}$ | Main engine conversion factor between fuel consumption and $CO_2$ emission. |
| $SFC_{ME}$ | Main engine specific fuel consumption. |

- Auxiliary Engines Emissions: Factor *B*

$$B = ( P_{AE}\, c_{F\,AE}\, SFC_{AE} ) \tag{A3}$$

| | |
|---|---|
| $P_{AE}$ | Power of auxiliary engines. |
| $c_{F\,AE}$ | Auxiliary engine conversion factor between fuel consumption and $CO_2$ emission. |
| $SFC_{AE}$ | Auxiliary engine specific fuel consumption. |

- Generators/Motors Emissions: Factor *C*

$$C = \left( \left( \prod_{j=1}^{m} f_j \sum_{i=1}^{n} P_{PTI(i)} - \sum_{i=1}^{n} f_{eff(i)}\, P_{AE\,eff(i)} \right) c_{F\,AE}\, SFC_{AE} \right) \tag{A4}$$

| | |
|---|---|
| $f_i$ | Correction factor for ship specific design elements. |
| $P_{PTI}$ | Power of shaft motor divided by the efficiency of shaft generator. |
| $f_{eff}$ | Availability factor of innovative energy efficiency technology. |
| $P_{AE\,eff}$ | Auxiliary power reduction due to individual technologies for electrical energy efficiency. |
| $c_{F\,AE}$ | Auxiliary engine conversion factor between fuel consumption and $CO_2$ emission. |
| $SFC_{AE}$ | Auxiliary engine specific fuel consumption. |

- Efficiency Technologies: Factor *D*

$$D = \sum_{i=1}^{n} f_{eff(i)}\, P_{eff(i)}\, c_{F\,ME}\, SFC_{ME} \tag{A5}$$

| | |
|---|---|
| $f_{eff}$ | Availability factor of innovative energy efficiency technology. |
| $P_{eff}$ | Output power of innovative mechanical energy efficient technology. |
| $C_{F\,ME}$ | Main engine conversion factor between fuel consumption and $CO_2$ emission. |
| $SFC_{ME}$ | Main engine specific fuel consumption. |

- Transport Work: Factor *E*

$$E = f_i\, f_c\, C_p\, V_{ref}\, f_w \tag{A6}$$

| | |
|---|---|
| $f_i$ | Capacity factor. |
| $f_c$ | Cubic capacity correction factor. |
| $C_p$ | Capacity or deadweight. |
| $V_{ref}$ | Ship speed. |
| $f_w$ | Weather factor. |

**Appendix B**

$$EEOI = \frac{\sum_j F_j\, c_{F(j)}}{m_c\, D} \tag{A7}$$

| | |
|---|---|
| $j$ | Fuel type. |
| $F_j$ | Mass of consumed fuel $j$. |
| $c_{F(j)}$ | Fuel mass to $CO_2$ mass conversion factor for fuel $j$. |
| $m_c$ | Cargo carried (tonnes). |
| $D$ | Distance in nautical miles corresponding to the cargo carried. |

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
