# Peer review of "Optimization of the Emissions Profile of a Marine Propulsion System Using a Shaft Generator with Optimum Tracking-Based Control Scheme"

_jmse, doi:10.3390/jmse8030221_

Round 1
Reviewer 1 Report
Line 4 words ''thermal machines'' to be replaced with ''thermal cycle''.
Line 6 words ''oxide sulphur’’ and ‘’oxide nitrogen’’ to be rewritten as ‘’sulphur oxide’’ and ‘’nitrogen oxide’’.
Line 24 page puntuaction is required to be corrected as per J. Mar. Sci. Eng. guidance.
Line 29 The sentence ‘’ The ship’s design power…’’ to be changed to ‘’ The ship’s power design…’’.
Line 67 To reformulate sentence: ‘’ Both definitions can be worked together when analysing the efficiency of a vessel because in order to improve its efficiency we need to know its current efficiency’’.
Line 75 To reformulate sentence: ‘’ Because fuel consumption generates emissions, any reduction of its consumptions leads to lower 76 level of emissions generation, which provides the baseline of considering the use of EEDI and EEOI 77 as means of evaluation of vessel efficiency’’.
Line 145 As per MAN hybrid propulsion systems are made with four stroke generator sets. The four stroke generator is compatible with PSMS units and gearbox can be made as it was described in the text. However, two stroke main propulsion unit ever needs a variable pitch propeller or complicated planetary gear system for transmission of the revolution to the PSMS unit. So this will need to be explained in the text.
Carried numerical analysis may stand as it is made from 5000 kW onwards and planetary gears may cover a synchronization range of PSMS in the upper running zones of the main engine. According to MAN catalogue part load of the main engine 7G80ME-C9.2 is from 58 – 72 RPM and it will be prudent that analysis covers that power range of the main engine as it is realistic to expect that running range of complicated planetary transmission gears.
For the lower and maneuvering running zones it is hard to expect that planetary transmission gear will cover it under normal production cost.
Line 286 Figures quality 7-14 to be improved.
Line 287 To reformulate sentence: ‘’This value represents the amount of CO2 288 emissions that by design this ship can generate and because is a 289 design value can be modified at design stages when the main and auxiliary machinery is selected 290 and allocated to it.’’
Line 293 The sentence: ‘’Operational emissions are of great value when trying to compare or establish an analysis of the design emissions. ‘’ is not required in the text.
Line 297. To reformulate sentence as it is not clear: ‘’ EEDI provides a fixed value at the design conditions yet EEOI can be used to evaluate the performance at every variation of engine load because considers the amount of fuel consumed no matter the velocity, although the consideration of using mass of cargo transported and distance navigated is just another representation of cargo capacity and ship’s speed respectively.’’
Line 306 Better explanation of the Figure 8. Figure 8 should be below the related text. What range is SCF 2%, please explain and add the related text.
Line 311 Better explanation of Figure 9. Figure 9 to be below related text.
Line 318 Better explanation of Figure 10. Figure 10 to be below related text.
Line 348 Figure 14 should be inside chapter 5.4.
Line 368 Better explanation of figure 17.
Line 372 To highlight results in the conclusion clearly and to explain difficulties in the low running main engine zones for running PSMS unit with fixed propeller configuration.
Line 373 The sentence: ‘’ Results presented, differentiating the performance of the control scheme of the shaft generator, 374 are quite significant when compared to not having a shaft generator installed.’’ is not required
Line 380 The word ‘’assertiveness’’ to be replaced with ‘’high reliability’’ through the text.
Author Response
Thanks for reviewing the manuscript “Optimization of the Emissions Profile of a Marine Pro- pulsion System using a Shaft Generator with optimum tracking based Control Scheme”. I appreciate the careful review and constructive suggestions made, and for the extremely helpful comments provided for the paper.
For the corresponding comments with our responses, please see the attachment.

Reviewer 2 Report
It is reviewer opinion that the article, whose interest and rilevance was appreciated, is not accettable for the publication without major revisions, including the optimization of the overall on-board energy conversion system (main engine and diesel generators).

Author Response
Thanks for reviewing the manuscript “Optimization of the Emissions Profile of a Marine Pro- pulsion System using a Shaft Generator with optimum tracking based Control Scheme”. I appreciate the careful review and constructive suggestions made, and for the extremely helpful comments provided for the paper.
For the corresponding comments with our responses please see the attachment.

Round 2
Reviewer 1 Report
Article should be accept for publishing.
Author Response
Thanks for taking a second review of the manuscript “Optimization of the Emissions Profile of a Marine Propulsion System using a Shaft Generator with optimum tracking based Control Scheme”. We appreciate the careful review and constructive suggestions made, and for the extremely helpful comments provided for the paper.
Reviewer 2 Report
Is reveiwer opinon that the article, although improved, is not acceptable for the publication without the required improvements.

Author Response
Thanks for taking a second review of the manuscript “Optimization of the Emissions Profile of a Marine Propulsion System using a Shaft Generator with optimum tracking based Control Scheme”. We appreciate the careful review and constructive suggestions made, and for the extremely helpful comments provided for the paper.
Please see the attachment.
